# OpenReview forum: "Accelerating first-order methods for nonconvex-strongly-convex bilevel optimization under general smoothness"
_NeurIPS.cc/2025/Conference — Submitted to NeurIPS 2025_

### Official Review · Reviewer_aCYi · 2025-06-26

**Clarity:** 3
**Significance:** 2
**Originality:** 2
**Rating:** 2
**Confidence:** 4

**Summary:**

This paper introduces an accelerated first-order framework for solving nonconvex-strongly-convex bilevel optimization problems, extending existing nonconvex optimization techniques to a broader setting with generalized Hölder continuity assumptions. The proposed approach utilizes accelerated gradient descent (AGD) combined with a carefully designed restart condition. When reduced to the standard Lipschitz continuity setting, the method achieves an improved oracle complexity of $\widetilde{O}(\ell^{3/4}\kappa^{13/4}\epsilon^{-7/4})$ without requiring access to second-order oracles. Experiments on data hypercleaning and hyperparameter optimization tasks validate the effectiveness of the proposed algorithm.

**Questions:**

- Why does the proposed method RAGD-GS exhibit slightly worse dependency on $\kappa$ compared to PRAF2BA [1]? Given that these two algorithms are quite similar, could you specify which technical part leads to this gap in the dependence on $\kappa$-dependency gap in Gc$(f, \epsilon)$? In addition, why does this gap not appear in Gc$(g, \epsilon)$?

- I suggest that the authors include comparisons with at least PRAF2BA [1] and other bilevel baselines discussed therein (see their Figures 1 and 2 in Appendix A) for both data hypercleaning and hyperparameter optimization experiments, as these two works study the same problem setting and such comparisons would enable a more fair evaluation.

- What is the motivation behind generalizing Assumption 1(v) from [1] to a broader Hölder continuity condition and introducing an additional Hölder continuity assumption on $\nabla\_{xx}f(x,y)$? Furthermore, why do you choose not to similarly generalize the other Lipschitz continuity assumptions to Hölder continuity?



[1] Li, Chris Junchi. "Accelerated Fully First-Order Methods for Bilevel and Minimax Optimization." arXiv preprint arXiv:2405.00914 (2024).

**Ethical Concerns:**

["NO or VERY MINOR ethics concerns only"]

**Final Justification:**

The authors' rebuttal addresses most of my concerns. However, two major weaknesses remain:

- First, the algorithm design and theoretical analysis largely build on the combination of ideas from [1, 2] and [3, 4]. While the paper introduces more general smoothness assumptions, the core analytical framework remains similar. This concern is also echoed by reviewer Kp5u in question 3.

- Second, the proposed algorithm and analysis are limited to the deterministic setting and do not extend to the stochastic setting, which restricts the practical applicability of the method. Reviewer 3FGu raised a similar concern in weakness 2.

Therefore, I maintain my rating.

[1] Yang, Haikuo, Luo Luo, Chris Junchi Li, and Michael I. Jordan. "Accelerating inexact hypergradient descent for bilevel optimization." arXiv preprint arXiv:2307.00126 (2023).

[2] Li, Chris Junchi. "Accelerated Fully First-Order Methods for Bilevel and Minimax Optimization." arXiv preprint arXiv:2405.00914 (2024).

[3] Kwon, Jeongyeol, Dohyun Kwon, Stephen Wright, and Robert D. Nowak. "A fully first-order method for stochastic bilevel optimization." In International Conference on Machine Learning, pp. 18083-18113. PMLR, 2023.

[4] Chen, Lesi, Yaohua Ma, and Jingzhao Zhang. "Near-optimal nonconvex-strongly-convex bilevel optimization with fully first-order oracles." Journal of Machine Learning Research 26, no. 109 (2025): 1-56.

**Limitations:**

Yes.

**Paper Formatting Concerns:**

The instruction block of the checklist was not deleted.

**Quality:**

2

**Strengths And Weaknesses:**

**Strengths**

- The paper is clearly written and easy to follow.

- This work extends techniques from nonconvex optimization to a broader setting by considering Hölder continuity assumptions on both the upper-level and lower-level objectives for solving nonconvex-strongly-convex bilevel optimization problems. In particular, under the standard Lipschitz continuity setting, the proposed method obtains an accelerated oracle complexity of $\widetilde{O}(\kappa^{13/4}\epsilon^{-7/4})$, relying solely on first-order oracle information (without access to second-order oracles).

- Experimental results validate theoretical findings.

**Weaknesses**

- There are many other relevant works in the field of bilevel optimization that have not been discussed in the related work section (Section 1.2); for example, see [1], [2], and references therein.

- The algorithm design lacks novelty. Specifically, the proposed RAGD-GS (Algorithm 1) is essentially identical to Algorithms 1 and 2 from [3], where accelerated gradient descent (AGD) combined with a restart technique was introduced to achieve an accelerated convergence rate of $\widetilde{O}(\kappa^{13/4}\epsilon^{-7/4})$ for bilevel optimization using fully first-order methods.

- It seems that the main difference between this paper and [3] lies in generalizing Assumption 1(v) from [3], which originally stated as Lipschitz continuity, to a more general Hölder continuity assumption, along with an additional Hölder continuity condition on $\nabla\_{xx}f(x,y)$.

- The experimental section on data hypercleaning and hyperparameter optimization should include additional baseline methods. At a minimum, it should include comparisons with PRAF2BA [3] and other baselines discussed therein (see their Appendix A). For a more fair and more comprehensive evaluation, the authors could also consider including representative bilevel baselines from [1] (see their Figures 3 and 4 in Appendix A).

- In Section 1.2, the authors claim that the proposed method improves the first-order oracle complexity for finding an $\epsilon$-stationary point to $\widetilde{O}(\kappa^{13/4}\epsilon^{-7/4})$. However, as presented in Table 1 of [3], PRAF2BA already achieves a complexity of $\widetilde{O}(\kappa^{11/4}\epsilon^{-7/4})$ for Gc$(f, \epsilon)$ and $\widetilde{O}(\kappa^{13/4}\epsilon^{-7/4})$ for Gc$(g, \epsilon)$, respectively. This shows that PRAF2BA has better dependency on the largest condition number $\kappa$ compared to RAGD-GS.


[1] Chen, Xuxing, Tesi Xiao, and Krishnakumar Balasubramanian. "Optimal algorithms for stochastic bilevel optimization under relaxed smoothness conditions." Journal of Machine Learning Research 25, no. 151 (2024): 1-51.

[2] Kwon, Jeongyeol, Dohyun Kwon, and Hanbaek Lyu. "On the complexity of first-order methods in stochastic bilevel optimization." arXiv preprint arXiv:2402.07101 (2024).

[3] Li, Chris Junchi. "Accelerated Fully First-Order Methods for Bilevel and Minimax Optimization." arXiv preprint arXiv:2405.00914 (2024).

---

> ### Author Rebuttal · Authors · 2025-07-31
>
> We thank Reviewer aCYi for the detailed and thoughtful review. We are grateful for the recognition that:
> - Our paper is **clearly written and easy to follow**;
> - We provide a **fully first-order accelerated method with provable complexity guarantees** under general Hölder smoothness, extending recent nonconvex optimization techniques to the bilevel setting;
> - Our **experiments validate the theoretical results**.
>
> We respectfully acknowledge the concerns raised regarding novelty and literature coverage. Below, we provide point-by-point clarifications and improvements to address these issues.
>
> ---
>
> ### W1. On related works in bilevel optimization not covered (e.g., [1], [2])
>
> We appreciate the reviewer’s suggestion and fully agree that [1] and [2] are highly relevant.
>
> [1] proposes the Moving-Average Stochastic Bilevel Optimization (MA-SOBA) framework, which achieves an optimal sample complexity of $\mathcal{O}(\epsilon^{-2})$ under relaxed smoothness assumptions.
> [2] studies the complexity of first-order methods in stochastic bilevel optimization, providing tight upper and lower bounds under different assumptions.
>
> These works address **stochastic** bilevel optimization, whereas our method focuses on **deterministic** problems under general Hölder continuity, using **fully first-order updates**. We will revise the related work section to incorporate a detailed comparison with [1, 2] in terms of oracle complexity, smoothness assumptions, and algorithmic distinctions.
>
> ---
>
> ### W2. Novelty beyond [3, "PRAF2BA"]
>
> We thank the reviewer for highlighting this. Our approach is distinguished from [3] in several key aspects:
>
> 1. **Restart scheme:** Our scheme (Eq. (5)) is tailored to **restricted Hölder continuity**, and becomes **independent of $\epsilon$** when $\nu_f = \nu_g = 1$. In contrast, [3] uses an $\epsilon$-dependent threshold.
>
> 2. **Descent analysis:** We introduce a novel potential function to handle errors in $\nabla L_\lambda^\star(x)$ (see Lemma 7), which is not present in [3].
>
> 3. **Weaker precision requirement:** Our method only requires $\sigma \propto \epsilon$, while [3] requires $\sigma \propto \epsilon^2$.
>
> 4. **Same convergence rate in Lipschitz case:** Under $\nu_f = \nu_g = 1$, our rate matches that of PRAF2BA: $\widetilde{\mathcal{O}}(\epsilon^{-7/4})$.
>
> 5. **Hölder-smooth complexity:** We are, to our knowledge, the first to achieve:
>    $$
>    \mathcal{O}\left(\Delta \ell^{\frac{2+2\nu_f-\nu_f\nu_g}{2+2\nu_f}} \kappa^{\frac{6+7\nu_f-2\nu_f\nu_g}{2+2\nu_f}} \epsilon^{-\frac{4+4\nu_f-\nu_f\nu_g}{2+2\nu_f}}\right)
>    $$
>    under general Hölder smoothness using only first-order information.
>
> We will highlight these contributions more clearly in the revised introduction and discussion sections.
>
> ---
>
> ### W3. Assumption
>
> Lipschitz assumptions on $f$ and $g$ are common in the bilevel literature (e.g., [3–5]). Our generalization to Hölder continuity enables the inclusion of more realistic and structured objectives without compromising tractability. See the response to Q3 for details. We will clarify this motivation in the revised assumptions section.
>
> ---
>
> ### W5 & Q1. $\kappa$ clarification
>
> Thank you for the opportunity to clarify. Our definition of $\kappa$ includes all relevant smoothness constants up to second order:
>
> $$
> \ell := \max\{C_f, L_f, \rho_f, H_f, L_g, \rho_g, M_g\}
> $$
>
> This generalization is aligned with [4]. In contrast, [3] uses a simplified $\kappa = \max\{L_f, L_g\}/\mu$. In practice, [3]'s convergence bounds depend on higher-order constants even though they are omitted.
>
> Accordingly, we refine Table 1 to use our definition of $\kappa$:
>
> | $ $              | Gc($f$, $\epsilon$)                          | Gc($g$, $\epsilon$)                          | JV($g$, $\epsilon$)                          | HV($g$, $\epsilon$)                          |
> |------------------|---------------------------------------------|---------------------------------------------|---------------------------------------------|---------------------------------------------|
> | RAHGD ([5])      | $\tilde{\mathcal{O}}(\ell^{3/4}\kappa^{11/4} \epsilon^{-7/4})$ | $\tilde{\mathcal{O}}(\ell^{3/4}\kappa^{13/4} \epsilon^{-7/4})$ | $\tilde{\mathcal{O}}(\ell^{3/4}\kappa^{11/4} \epsilon^{-7/4})$ | $\tilde{\mathcal{O}}(\ell^{3/4}\kappa^{13/4} \epsilon^{-7/4})$ |
> | PRAF2BA ([3])    | $\tilde{\mathcal{O}}(\ell^{3/4}\kappa^{13/4} \epsilon^{-7/4})$ | $\tilde{\mathcal{O}}(\ell^{3/4}\kappa^{13/4} \epsilon^{-7/4})$ | \| \| | \| \| |
>
> This ensures consistency and transparency in how $\kappa$ is used across methods.
>
> ---
>
> ### W4 & Q2. On including comparisons with PRAF2BA and other baselines
>
> We agree that including PRAF2BA is essential. Due to space limits, we included results on the Data Cleaning task ($p=0.2$). Other baselines such as AID-BIO, BA-CG, RAHGD, and ITD-BIO were also aligned with those used in [3].
>
> | training loss | time |
> |---------------|------|
> | 19.03456      | 0 |
> | 3.276573      | 7.089338 |
> | 2.417393      | 13.98089 |
> | 2.351485      | 20.73936 |
> | 2.348124      | 27.54747 |
> | 2.344714      | 34.40922 |
> | 2.318441      | 41.28116 |
> | 2.334415      | 48.27428 |
> | 2.32698       | 55.20042 |
> | 2.358314      | 62.06065 |
> | 2.347294      | 69.05019 |
> | 2.334028      | 75.96848 |
> | 2.325851      | 82.93495 |
> | 2.366283      | 89.99217 |
> | 2.352741      | 97.08053 |
> | 2.336671      | 104.0516 |
> | 2.347916      | 110.9669 |
> | 2.343958      | 117.8462 |
> | 2.353041      | 124.7844 |
> | 2.345995      | 131.6605 |
> | 2.346084      | 138.7054 |
> | 2.33633       | 145.5454 |
> | 2.33195       | 152.2752 |
> | 2.31613       | 159.1535 |
> | 2.322544      | 165.932 |
> ---
> ### Q3. On motivation for Hölder continuity generalization
> We appreciate this thoughtful question. Our motivation for Hölder continuity generalization is threefold:
> 1. Theoretical Generalization: Hölder continuity is a strict generalization of Lipschitz continuity, allowing us to cover a broader class of functions. For example, $f(x) = \sqrt{x}$ is (0.5,1)-Hölder continuous but not Lipschitz continuous. Moreover, the optimization community has long studied such settings from a theoretical perspective [6–11].
> 2. Emerging Applications: Recent studies in related areas, such as positional encodings in graph learning, have demonstrated Hölder continuity properties [11]. These suggest that such regularity conditions may naturally arise in structured models or neural network components. This opens the door to identifying or designing architectures where Hölder-type conditions are more appropriate.
> 3. Timeliness and Potential Impact: As evidenced in works like [10], there is growing interest in developing optimization algorithms under relaxed smoothness assumptions. By establishing provable guarantees for bilevel optimization under Hölder continuity, we aim to contribute foundational insights that may inspire further investigation into both theoretical properties and practical relevance.
> Our decision to generalize only Assumption 1(v)  from [1](i.e., the Hölder continuity of $\nabla^2\_{xx} f$) was motivated by both analytical tractability and practical relevance. Specifically, this term governs the curvature of the upper-level objective with respect to the decision variable $x$, which plays a central role in our the property of $\nabla^2 L\_\lambda^*(x)$.
> On the other hand, extending the Hölder continuity to cross-derivatives such as $\nabla^2\_{xy} f$ introduces significant technical challenges. In particular, under Hölder continuity of $\nabla^2\_{xy} f$, it becomes difficult to tightly control terms like $||\nabla^2\_{xy} f||$ in Lemma 2, which would be necessary to ensure that the gradient approximation error remains bounded.
> Therefore, to maintain clarity and ensure provable convergence under the general Hölder setting, we chose to relax only the most influential smoothness assumption—while keeping others Lipschitz continuous to avoid overly technical complications. We leave further generalizations as future work.
> ---
>
> ### Summary
>
> We sincerely thank the reviewer for their thoughtful and detailed feedback. In the final version, we will (i) expand the related work to include [1,2], (ii) clarify our technical contributions beyond [3], and (iii) include PRAF2BA baselines and revise our complexity table accordingly. We hope these clarifications help demonstrate the novelty and rigor of our work.
>
> ---
> **References**
>
> [1] Chen et al, "Optimal algorithms for stochastic bilevel optimization under relaxed smoothness conditions." Journal of Machine Learning Research 25, no. 151 (2024): 1-51.
> [2] Kwon et al, "On the complexity of first-order methods in stochastic bilevel optimization." arXiv preprint arXiv:2402.07101 (2024).
> [3] Li,  "Accelerated Fully First-Order Methods for Bilevel and Minimax Optimization." arXiv preprint arXiv:2405.00914 (2024).
> [4] Chen et al, "Near-optimal nonconvex-strongly-convex bilevel optimization with fully first-order oracles." arXiv preprint arXiv:2306.14853 (2023).
> [5] Yang et al, "Accelerating inexact hypergradient descent for bilevel optimization." arXiv preprint arXiv:2307.00126 (2023).
> [6] Berger et al, "On the quality of first-order approximation of functions with Hölder continuous gradient", JOTA 2020.
> [7] Nesterov, "Universal gradient methods for convex optimization problems", Mathematical Programming 2015.
> [8] Lan, Bundle-level type methods uniformly optimal for smooth and nonsmooth convex optimization, Mathematical Programming 2015.
> [9] Chien et al, Convergent Privacy Loss of Noisy-SGD without Convexity and Smoothness, ICLR 2025.
> [10] Marumo et al, "Universal heavy-ball method for nonconvex optimization under  hölder continuous hessians". Mathematical Programming, pages 1–29(2024).
> [11] Huang et al., "On the Stability of Expressive Positional Encodings for Graphs", ICLR (2024).

---

> > ### Comment · Reviewer_aCYi · 2025-08-02
> >
> > I thank the authors for their thorough rebuttal. Regarding the third point in response to Q3, could the authors clarify what Assumption 1(v) in [1] refers to? I was unable to locate it in [1] and wonder if there may be a typo.
> >
> > In addition, is it possible to extend the current work and proposed algorithm to the stochastic setting? This concern was also raised by reviewer 3FGu. Limiting the analysis to the deterministic case may restrict the applicability of the method in practice.
> >
> > [1] Chen, Xuxing, Tesi Xiao, and Krishnakumar Balasubramanian. "Optimal algorithms for stochastic bilevel optimization under relaxed smoothness conditions." Journal of Machine Learning Research 25, no. 151 (2024): 1-51.

---

> > > ### Author Response · Authors · 2025-08-04
> > > **Reply to Reviewer aCYi**
> > >
> > > We sincerely thank the reviewer for their careful reading and thoughtful follow-up questions.
> > >
> > > **Clarification on Assumption Reference**:
> > > We apologize for the confusion regarding the reference to "Assumption 1(v) in [1]." This was a typographical error. The intended reference was to *Assumption 1(v) in [2]*, but not [1]. We will revise the text to clearly distinguish between our assumptions and those in the cited references.
> > >
> > > **Extension to the Stochastic Setting**:
> > > We greatly appreciate the reviewer—along with reviewer 3FGu—for raising the question regarding the extension of our approach to the stochastic setting. Fang et al. [3] provide a sharp analysis demonstrating that SGD can find an $(\epsilon, O(\epsilon^{0.5}))$-approximate second-order stationary point in $\tilde{\mathcal{O}}(\epsilon^{-3.5})$ stochastic gradient computations, assuming gradient- and Hessian-Lipschitz conditions with dispersive noise. While these results are promising, they are specific to single-level settings. Incorporating acceleration and Hölder-type smoothness into stochastic bilevel optimization poses several non-trivial challenges:
> > >
> > > 1. Restart Condition Design: Our accelerated algorithm relies on a deterministic restart mechanism (see Eq. (5)) to achieve improved convergence. In stochastic settings, quantities used to trigger restarts (e.g., gradient magnitudes or potential decrease) are inherently noisy, making it difficult to determine when restarts should occur.
> > > 2. Gradient Precision Requirements: The theoretical complexity of our method depends on achieving inner-loop gradient estimators with precision $\sigma \propto \epsilon$. In stochastic bilevel settings, such precision typically requires large batch sizes or sample averages, especially when dealing with noisy or biased hyper-gradients. This can significantly increase sample complexity and complicate convergence analysis.
> > >
> > > To the best of our knowledge, there is currently no existing first-order stochastic bilevel method that simultaneously incorporates acceleration via restart schemes and handles generalized smoothness assumptions such as Hölder continuity. Developing such a unified framework remains an open and challenging problem.
> > >
> > > We believe that our current work can serve as a theoretical stepping stone, and we view stochastic generalization as a promising avenue for future work.  Once again, we are grateful for your valuable comments and suggestions.
> > >
> > >
> > > ---
> > > References
> > >
> > > [1] Chen, Xuxing, Tesi Xiao, and Krishnakumar Balasubramanian. "Optimal algorithms for stochastic bilevel optimization under relaxed smoothness conditions." Journal of Machine Learning Research 25, no. 151 (2024): 1-51.
> > >
> > > [2] Li, Chris Junchi. "Accelerated Fully First-Order Methods for Bilevel and Minimax Optimization." arXiv preprint arXiv:2405.00914 (2024).
> > >
> > > [3] Fang, Cong, Zhouchen Lin, and Tong Zhang. Sharp Analysis for Nonconvex SGD Escaping from Saddle Points. In Proceedings of the Thirty-Second Conference on Learning Theory (COLT), pages 1192–1234. PMLR(2019).

---

> > > > ### Comment · Reviewer_aCYi · 2025-08-09
> > > >
> > > > Thank you again for the detailed response. The reference now looks good to me. I encourage the authors to explore the stochastic bilevel setting to further strengthen the manuscript.

---

### Official Review · Reviewer_Kp5u · 2025-07-01

**Clarity:** 3
**Significance:** 3
**Originality:** 2
**Rating:** 4
**Confidence:** 3

**Summary:**

This paper presents an accelerated first-order method for solving nonconvex–strongly-convex bilevel optimization problems. The authors combine a restarting technique with penalty methods to achieve a faster convergence rate, with $\mathcal{O}(\epsilon^{7/4})$ iteration complexity for finding an $\epsilon$-stationary point.
Their algorithms and convergence guarantees can be extended to more general cases—going beyond Lipschitz smoothness to Hölder continuity of gradients and Hessians.
Numerical results also demonstrate good performance.

**Questions:**

1. The main concern is novelty. [Li and Lin, 2023] propose an accelerated first-order algorithm for nonconvex problems with Hölder smoothness, and [Kwon et al., 2023; Chen et al., 2023] present fully first-order algorithms for solving nonconvex–strongly-convex bilevel optimization.
This paper merely combines the algorithms and theoretical analyses from those works to solve bilevel optimization with a faster rate under more general smoothness assumptions. Obtaining such results by combining the techniques in these papers is not surprising. The authors should further clarify the novelty.

	2.	More discussion is needed on why Hölder continuity should be considered in bilevel optimization. The authors should provide additional examples where Lipschitz smoothness fails but Hölder smoothness holds.

	3.	Because the theoretical results are given for the gradient norm, I suggest that the authors provide experiments demonstrating convergence of the gradient norm, perhaps using some toy examples.

**Ethical Concerns:**

["NO or VERY MINOR ethics concerns only"]

**Final Justification:**

Based on the paper and the authors’ rebuttal, I agree that the extension from single-level to bilevel is nontrivial, so I am raising the score to 4.

**Quality:**

2

**Strengths And Weaknesses:**

Strengths:

	1.	The method is fully first-order and achieves a faster convergence rate than the benchmark methods in [Chen et al., 2023] and [Kwon et al., 2023].

	2.	The analysis can be extended to more general Hölder-continuity settings.

Weaknesses: see Questions.

---

> ### Author Rebuttal · Authors · 2025-07-31
>
> We thank reviewer Kp5u for their constructive comments and for recognizing key strengths of our work, including the **fully first-order nature** of the proposed algorithm and its **faster convergence rate** compared to prior first-order methods such as [7] and [10]. Below, we address the reviewer’s concerns in detail.
>
> ---
>
> ### Q1. On the Novelty and Contribution Beyond Prior Works
>
> We appreciate the opportunity to clarify the novelty of our approach. While our work is indeed inspired by ideas from [8], [7], and [10], it is **not a mere combination** of existing techniques. Our contributions introduce several **key innovations**, both algorithmic and analytical, which we summarize below:
>
> 1. **Extension of acceleration to bilevel problems under general smoothness assumptions**:
>    - In contrast to [9], which addresses single-level nonconvex optimization, our work targets the **more challenging bilevel setting**.
>    - We generalize smoothness assumptions and accelerate first-order methods beyond the frameworks in [7] and [10].
>
> 2. **New algorithmic design for bilevel acceleration**:
>    - Our restart scheme (see Eq. (5)) differs from that in [9], as it is designed to operate under **restricted Hölder continuity**.
>      Notably, when $\nu_f = \nu_g = 1$, our restart threshold becomes **independent of** $\epsilon$, whereas [9] uses a threshold that explicitly depends on $\epsilon$.
>    - We introduce a **potential function analysis** that carefully accounts for gradient estimation errors in $\nabla L\_\lambda^\star(x)$ (see Lemma 7).
>    - Our analysis allows the inner-loop solvers to compute approximate gradients with **precision $\sigma \propto \epsilon$**, in contrast to [8], which requires **$\sigma \propto \epsilon^2$**.
>
> 3. **Complexity improvements using only first-order information**:
>    - Compared to [7], our method achieves a **faster convergence rate** of $\widetilde{O}(\epsilon^{-7/4})$.
>    - While [8] achieves the same $\epsilon$-dependence, it relies on **second-order information** (Hessian-vector products), whereas our method uses **only gradients**.
>
> We will explicitly summarize these technical distinctions in the revised version, particularly in the introduction and discussion sections.
>
> ---
>
> ### Q2. Motivation for Hölder Continuity in Bilevel Optimization
>
> We appreciate this insightful question. While Lipschitz smoothness is a common assumption, it can be **overly restrictive** in practice, especially in bilevel settings with complex or structured objectives.
>
> Our motivation for considering **Hölder continuity** is threefold:
>
> 1. **Theoretical generalization**:
>    Hölder continuity strictly generalizes Lipschitz smoothness. For example, $f(x) = \sqrt{x}$ is $(0.5, 1)$-Hölder continuous but not Lipschitz continuous. Theoretical works have long studied optimization under such assumptions [1–5].
>
> 2. **Emerging applications**:
>    Recent studies in graph learning have identified Hölder-type behavior in positional encodings [6]. While these do not directly correspond to gradient Hölder continuity, they suggest that such regularity may naturally arise in structured models or neural networks.
>
> 3. **Timeliness and future potential**:
>    The optimization community is actively exploring relaxed smoothness assumptions (e.g., [5]), indicating a trend toward broadening the theoretical foundation of optimization algorithms. Our work provides **provable guarantees** under Hölder continuity for bilevel problems, paving the way for future exploration of this setting.
>
> In summary, Hölder continuity allows us to study a **broader class of bilevel problems** beyond standard smoothness. We view our work as an important step toward a more general theory of bilevel optimization.
>
> ---
>
> ### Q3. On Experiments Showing Gradient Norm Convergence
>
> We agree that convergence in $\|\|\nabla \varphi(x)\|\|$ (or the surrogate $\|\|\nabla L^\star_\lambda(x)\|\|$) is an important empirical indicator to support the theory.
> We have already computed these gradient norms during training in the **Data Cleaning** experiment. The table below summarizes the results for two noise levels ($p = 0.2$ and $p = 0.4$), showing clear convergence toward stationarity:
>
> | Epoch | $\|\|\nabla \varphi(x)\|\|$ ($p = 0.2$) | $\|\|\nabla \varphi(x)\|\|$ ($p = 0.4$) |
> |-------|---------------------------|---------------------------|
> | 0     | 0.2639                    | 0.2400                    |
> | 1     | 0.0078                    | 0.0079                    |
> | 2     | 0.0073                    | 0.0072                    |
> | 3     | 0.0074                    | 0.0073                    |
> | 4     | 0.0071                    | 0.0074                    |
> | 5     | 0.0070                    | 0.0071                    |
> | 6     | 0.0070                    | 0.0069                    |
> | 7     | 0.0069                    | 0.0068                    |
> | 8     | 0.0069                    | 0.0067                    |
> | 9     | 0.0068                    | 0.0067                    |
>
> We will include these results and their visualizations in the final version to better support the theoretical claims.
>
> ---
>
> ### Summary
>
> We thank the reviewer for highlighting important points around novelty, smoothness assumptions, and empirical validation. In response, we will:
>
> - Clearly articulate our technical contributions beyond prior work;
> - Expand the discussion of Hölder continuity and its relevance;
> - Include convergence plots for gradient norm to support theoretical guarantees.
>
> We hope these clarifications demonstrate the **significance, rigor, and originality** of our work.
>
> ---
>
> **References**
> [1] Berger et al. *On the quality of first-order approximation of functions with Hölder continuous gradient*. JOTA, 2020.
> [2] Nesterov. *Universal gradient methods for convex optimization problems*. Mathematical Programming, 2015.
> [3] Lan. *Bundle-level methods for smooth and nonsmooth convex optimization*. Math Programming, 2015.
> [4] Chien et al. *Convergent Privacy Loss of Noisy-SGD without Convexity and Smoothness*. ICLR, 2025.
> [5] Marumo & Takeda. *Universal heavy-ball method under Hölder continuous Hessians*. Mathematical Programming, 2024.
> [6] Huang et al. *On the Stability of Expressive Positional Encodings for Graphs*. ICLR, 2024.
> [7] Chen et al. *Near-optimal nonconvex-strongly-convex bilevel optimization with fully first-order oracles*. arXiv, 2023.
> [8] Yang et al. *Accelerating inexact hypergradient descent for bilevel optimization*. arXiv, 2023.
> [9] Li et al. *Restarted Nonconvex Accelerated Gradient Descent: No More Polylogarithmic Factor in the $O(\epsilon^{-7/4})$ Complexity*. PMLR, 2022.
> [10] Kwon et al. *A fully first-order method for stochastic bilevel optimization*. ICML, 2023.

---

> > ### Comment · Reviewer_Kp5u · 2025-08-07
> >
> > Thank you, the authors have addressed my concern. I will raise the score to 4.

---

> ### Author Response · Authors · 2025-08-08
>
> Dear Reviewer Kp5u,
>
> We hope this message finds you well.
>
> We truly appreciate your earlier indication that you might consider increasing your score, and we would be very grateful if you feel our responses have resolved the issues and could kindly reflect this in the system (e.g. via the acknowledgement / final justification).
>
> If there remain any points that are unclear or that you would like us to expand on, we would be glad to provide further clarification at your convenience.
>
> Thank you again for your time and constructive feedback! It has been very helpful to us.
>
> Best regards,
>
>
> The Authors

---

### Official Review · Reviewer_bRp6 · 2025-07-02

**Clarity:** 1
**Significance:** 3
**Originality:** 3
**Rating:** 4
**Confidence:** 3

**Summary:**

This paper introduces an accelerated first-order method (RAGD-GS) for solving nonconvex–strongly-convex bilevel optimization problems under general smoothness conditions, including Hölder continuity of the derivatives of the outer and inner level objective functions. The authors extend recent advances in accelerated gradient methods for nonconvex optimization to the bilevel setting, propose a novel restart condition independent of $\epsilon$, and derive improved oracle complexity guarantees. Their method achieves an $O(\eps^{-7/4})$ convergence rate using only first-order information. The convergence analysis relies on the assumption that the inner-loop solvers produce approximate gradients up to a specified precision $\sigma$ proportional to $\epsilon$.
The paper includes theoretical analysis, and presents empirical results on hyperparameter optimization and data hypercleaning tasks

**Questions:**

- Clarify the trade-off in Lemma 2
The bound in Lemma 2 includes two terms, one increasing with ν_f and the other decreasing. Please provide intuition or discussion about this trade-off. Why does one term behave oppositely to the other? What does this mean for algorithm design or hyperparameter choices?

- Explain the role of technical lemmas
Currently, the paper reads like a sequence of inequalities. Please explain how Lemmas 5, 6, 8 contribute to the overall convergence proof. For example:
Why do we need to control the error in averaged gradients (Lemma 5)?
Under what conditions does Lemma 8 guarantee descent?

- Clarify inconsistencies in Lemma 8 discussion (l192):
You mention Lemma 8 as if it requires exact gradients, but it is explicitly stated under Condition 1 (inexact gradients). Please clarify this point or revise the sentence for consistency.

- Better explain the superiority Claim over Yang et al. [2023]:
On line 214, the paper states that RAGD-GS is “better than” prior work. This needs clarification. In what sense is it better — epsilon-dependence,  constants, use of first/second order information ?

**Ethical Concerns:**

["NO or VERY MINOR ethics concerns only"]

**Final Justification:**

See my answer to the authors and their answer to Q4.
The main contribution is to provide an algorithm that improves about the scheme of Yang et al in several (maybe incremental for some) aspects, eg order (first/second) of the information, smoothness, theoretical precision of the inner loop.

Note that Yang et al has only been published as a neurips 2023 workshop poster but Chen et al 2023 (mentioned in the rebuttal to Reviewer 3FGu and also cited in the paper) was published at JMLR.

So I think the result is of interest for the Neurips community.

Where I have a doubt is whether these improvements are rather weak or sufficient as a contribution.

**Limitations:**

The authors did not discuss the limitations of their work.

**Quality:**

2

**Strengths And Weaknesses:**

Strengths: The paper addresses a meaningful gap: achieving accelerated convergence rates for bilevel problems using only first-order information and under generalized smoothness conditions. The complexity results matches prior work (e.g., Yang et al. 2023) without second-order information.


Weaknesses:

1) Clarity:

The presentation suffers from a lack of clarity in key areas. Many of the technical lemmas (e.g., Lemmas 2, 5, 8), as well as the main algorithm are introduced with none or minimal explanation or intuition.This makes the narrative feel like a sequence of inequalities that are difficult to digest rather than a coherent argument.
For example:
- Lemma 2 introduces a bound whose structure is not explained — one term increases with ν_f, the other decreases — yet no intuition or interpretation is provided.
- The algorithm (Algorithm 1) could benefit from being rewritten more clearly, with explicit commentary on what each line accomplishes. In particular l167 the notation nabla L hat is not introduced (the estimator of the gradient) as the output of the loop in Algorithm 1.
- The roles of key lemmas (e.g., Lemma 5 controlling averaging error, Lemma 8 controlling descent) are not explained clearly, which obscures their importance in the convergence proof.
- line 192 after Lemma 8 is confusing: the authors claim “Lemma 8 shows that, if we use exact gradient ...” even though Lemma 8 assumes inexact gradients via Condition 1. I found this misleading.

2) Experiments:
Experimental comparisons are a bit limited; RAGD-GS performs similarly to RAHGD (Yang et al 2023) on small-scale tasks.

---

> ### Author Rebuttal · Authors · 2025-07-31
>
> We thank reviewer bRp6 for the thoughtful review and constructive suggestions. We are encouraged that the reviewer acknowledges the **importance of accelerating bilevel optimization using only first-order information** and the **generality of our Hölder-type smoothness assumptions**. Below, we address each concern in detail.
>
> ---
>
> ## W1. Clarity and Interpretation of Lemmas
>
> We appreciate the reviewer’s feedback regarding the lack of clarity in the presentation. We will substantially revise the narrative in the final version. Here, we clarify the reviewer’s questions point by point:
>
> #### (Q1) Lemma 2 – Interpretation of the trade-off:
>
> Lemma 2 provides a Hölder-type bound that consists of two terms:
> - The first term, $O(\ell \kappa^{\nu_f})$, **increases with $\nu_f$**, and reflects the contribution from the upper-level Hölder continuity;
> - The second term, $O(\lambda^{1-\nu_g} \ell \kappa^{4+\nu_g}) R^{1-\nu_f}$, **decreases with $\nu_f$**, and stems from the coupling between the upper and lower objectives via the penalty formulation (see Equation (16)).
>
> To provide more intuition: in the proof of Lemma 2, we decompose $\nabla^2 L_\lambda^\star(x)$ as:
> $$
> \nabla^2 L_\lambda^\star(x) = A(x) + B(x),
> $$
> where $A(x)$ corresponds to the Hessian of the penalized upper-level objective, and $B(x)$ arises from the perturbation of the lower-level optimal solution $y_\lambda^\star(x)$. The two terms yield:
> - $O(\ell \kappa^{\nu_f})$ (independent of $\lambda$),
> - $O(\lambda^{1-\nu_g} \ell \kappa^{4+\nu_g}) R^{1-\nu_f}$ (dependent on $\lambda$ unless $\nu_g = 1$).
>
> When $\nu_f$ decreases, the first term becomes smaller, as it relates to the Hölder constant of the composition of $\nabla_{xx}^2 f(x, y)$ with $y_\lambda^\star(x)$. However, the second term increases due to sensitivity to the radius $R$:
> $$
> \|x_1 - x_2\| = \|x_1 - x_2\|^{\nu_f} \cdot \|x_1 - x_2\|^{1 - \nu_f} \le R^{1 - \nu_f} \|x_1 - x_2\|^{\nu_f}.
> $$
> In Algorithm 1, we set $\mathcal{R}$ (Eq. (6)) and $\lambda$ (Theorem 1) such that the second term dominates. We will emphasize this intuition more clearly in the revised version.
>
> ---
>
> #### (Q2) Role of Lemmas 5, 6, and 8:
>
>   - Lemma 5 bounds the error between the average gradient over an epoch and the true gradient at the averaged iterate $\bar{w}_k$. This is essential because our algorithm outputs $\bar{w}_k$ and we need this estimate to control $\|\|\nabla L\_\lambda^\star(\bar{w}_k)\|\|$.
>   - Lemma 6 provides a Hessian-free quadratic surrogate for the function difference, used to show descent in the potential function (Eq. (9)).
>   - Lemma 8 gives the aggregate descent over one epoch. Together with Lemma 9, it ensures a convergence rate on $\min_i \|\|\nabla L^\star_\lambda(\bar{w}_i)\|\|$.
>
> We agree that the current version lacks intuitive discussion. In the final version, we will add brief commentary before each lemma to clarify its role in the convergence proof.
>
> ---
> ## W1. When does Lemma 8 ensure descent?
>
> As shown in the proof of Proposition 1, whenever $\bar{w}\_{t,k}$ is not an $\epsilon$-stationary point, Lemma 8 ensures that $L\_\lambda^\star(x)$ decreases in the t-th epoch.
>
> ---
>
> ## W1. Algorithm Clarity
>
> Thank you for the helpful suggestion. In the final version, we will:
> - Add line-by-line comments to Algorithm 1 to explain the purpose of each operation (e.g., hypergradient estimation, extrapolation, restart check);
> - Explicitly define the estimator $\hat \nabla L^\star_\lambda$ in the main text (used in Line 6), where:
> $$
> \hat \nabla L_\lambda^*(w_{t,k}) = \nabla_x f(w_{t,k}, y_{t,k}) + \lambda \left[\nabla_x g(w_{t,k}, y_{t,k}) - \nabla_x g(w_{t,k}, z_{t,k})\right],
> $$
> with $y_{t,k}$ and $z_{t,k}$ computed in Lines 4 and 5.
>
> ---
> ## Q3. Clarification on line 192 after Lemma 8:
>
> The sentence was poorly phrased. Lemma 8 holds for any $\sigma \geq 0$, where $\sigma$ quantifies the inaccuracy of the gradient estimator.
>
> To aid intuition: when $\sigma = 0$, i.e., the estimator is exact, the function value $L_\lambda^\star(x)$ decreases monotonically. This implies that **for sufficiently small $\sigma$**, the descent property remains approximately valid.
>
> Importantly, Algorithm 1 does **not** require $\sigma = 0$—convergence is guaranteed under bounded inexactness. We will revise the phrasing for clarity.
>
>
> ---
>
> ## Q4. Clarification on Comparison to Yang et al. (2023)
>
> The statement on line 214 that our method is "better than" [Yang et al., 2023] refers to three key distinctions:
>
> - Under standard Lipschitz continuity, we achieve the **same** convergence rate of $O(\epsilon^{-7/4})$ using **only first-order information**, while [Yang et al.] requires **Hessian-vector products**;
> - Under general Hölder continuity, we provide the **first provable acceleration result** for nonconvex–strongly convex bilevel problems;
> - Our analysis assumes that inner-loop solvers achieve gradient estimates with precision $\sigma = O(\epsilon)$, while [Yang et al.] and [Li et al.] require stricter precision $\sigma = O(\epsilon^2)$.
>
> We will clarify this sentence to better reflect our **advantages in oracle efficiency and smoothness generality**.
>
> ---
>
> ## W1. Experimental Comparison
>
> We agree that RAGD-GS and RAHGD exhibit similar performance on small-scale tasks, which is expected since both achieve the same asymptotic rate $\widetilde{O}(\epsilon^{-7/4})$. Our empirical contribution is to show that **RAGD-GS reaches this performance using only first-order gradients**, while RAHGD requires second-order information. We provide additional quantitative results of Data Hypercleaning over five different random seeds to strengthen this point.
>
> | Algorithm | p =0.2 (train loss) | p =  0.2 (Running Time, mean $\pm$ std) | p = 0.4 (train loss) | p =0.4 (Running Time, mean $\pm$ std)) |
> |---|---|---|---|---|
> | **RAGD-GS** |19.035| 0$\pm$0|19.035| 0$\pm$0|
> || 2.481 | 7.267 $\pm$ 0.300 | 2.498 | 7.530 $\pm$ 0.329 |
> | | 2.370 | 14.795 $\pm$ 0.737 | 2.365 | 15.142 $\pm$ 0.726 |
> | | 2.363 | 22.279 $\pm$ 1.043 | 2.360 | 22.813 $\pm$ 1.150 |
> | | 2.326 | 29.757 $\pm$ 1.425 | 2.348 | 30.506 $\pm$ 1.559 |
> | | 2.352 | 37.322 $\pm$ 1.857 | 2.336 | 38.144 $\pm$ 1.910 |
> | | 2.340 | 44.875 $\pm$ 2.221 | 2.312 | 45.774 $\pm$ 2.293 |
> | | 2.379 | 52.352 $\pm$ 2.607 | 2.324 | 53.355 $\pm$ 2.612 |
> | | 2.313 | 59.811 $\pm$ 2.929 | 2.355 | 61.086 $\pm$ 2.952 |
> | | 2.345 | 67.254 $\pm$ 3.304 | 2.318 | 68.674 $\pm$ 3.317 |
> | | 2.320 | 74.777 $\pm$ 3.669 | 2.344 | 76.266 $\pm$ 3.668 |
> | | 2.345 | 82.236 $\pm$ 4.048 | 2.311 | 83.947 $\pm$ 4.028 |
> | | 2.363 | 89.775 $\pm$ 4.380 | 2.325 | 91.692 $\pm$ 4.317 |
> | | 2.319 | 97.373 $\pm$ 4.677 | 2.342 | 99.348 $\pm$ 4.586 |
> | | 2.343 | 105.049 $\pm$ 5.002 | 2.318 | 106.943 $\pm$ 4.939 |
> | | 2.339 | 112.495 $\pm$ 5.129 | 2.334 | 114.558 $\pm$ 5.286 |
> | | 2.357 | 120.020 $\pm$ 5.300 | 2.332 | 122.203 $\pm$ 5.693 |
> | | 2.353 | 127.511 $\pm$ 5.410 | 2.318 | 129.776 $\pm$ 5.982 |
> | | 2.340 | 135.009 $\pm$ 5.479 | 2.323 | 137.505 $\pm$ 6.313 |
> | | 2.333 | 142.509 $\pm$ 5.575 | 2.326 | 145.200 $\pm$ 6.711 |
> | | 2.306 | 149.887 $\pm$ 5.774 | 2.283 | 152.876 $\pm$ 7.043 |
> | | 2.315 | 157.318 $\pm$ 6.016 | 2.339 | 160.518 $\pm$ 7.363 |
> | **RAHGD** |19.035| 0$\pm$0|19.035| 0$\pm$0|
> || 3.215 | 5.238 $\pm$ 0.054 | 3.277 | 5.312 $\pm$ 0.066 |
> | | 2.469 | 10.463 $\pm$ 0.083 | 2.436 | 10.502 $\pm$ 0.112 |
> | | 2.404 | 15.692 $\pm$ 0.081 | 2.383 | 15.714 $\pm$ 0.140 |
> | | 2.350 | 20.893 $\pm$ 0.090 | 2.361 | 20.946 $\pm$ 0.158 |
> | | 2.385 | 26.244 $\pm$ 0.063 | 2.355 | 26.148 $\pm$ 0.168 |
> | | 2.375 | 31.493 $\pm$ 0.064 | 2.325 | 31.407 $\pm$ 0.227 |
> | | 2.417 | 36.757 $\pm$ 0.118 | 2.331 | 36.689 $\pm$ 0.227 |
> | | 2.345 | 42.022 $\pm$ 0.163 | 2.370 | 41.977 $\pm$ 0.249 |
> | | 2.375 | 47.243 $\pm$ 0.223 | 2.330 | 47.183 $\pm$ 0.297 |
> | | 2.345 | 52.511 $\pm$ 0.273 | 2.365 | 52.374 $\pm$ 0.305 |
> | | 2.375 | 57.832 $\pm$ 0.327 | 2.328 | 57.587 $\pm$ 0.325 |
> | | 2.403 | 63.160 $\pm$ 0.355 | 2.339 | 62.804 $\pm$ 0.307 |
> | | 2.351 | 68.480 $\pm$ 0.519 | 2.362 | 68.005 $\pm$ 0.295 |
> | | 2.383 | 73.844 $\pm$ 0.474 | 2.338 | 73.289 $\pm$ 0.269 |
> | | 2.376 | 79.162 $\pm$ 0.513 | 2.351 | 78.672 $\pm$ 0.209 |
> | | 2.387 | 84.456 $\pm$ 0.480 | 2.352 | 83.915 $\pm$ 0.221 |
> | | 2.386 | 89.692 $\pm$ 0.506 | 2.340 | 89.101 $\pm$ 0.244 |
> | | 2.374 | 94.928 $\pm$ 0.424 | 2.339 | 94.354 $\pm$ 0.205 |
> | | 2.367 | 100.269 $\pm$ 0.291 | 2.339 | 99.634 $\pm$ 0.239 |
> | | 2.342 | 105.533 $\pm$ 0.295 | 2.292 | 105.029 $\pm$ 0.179 |
> | | 2.348 | 110.828 $\pm$ 0.272 | 2.360 | 110.262 $\pm$ 0.149 |
> | | 2.394 | 116.088 $\pm$ 0.272 | 2.339 | 115.506 $\pm$ 0.141 |
> | | 2.376 | 121.354 $\pm$ 0.284 | 2.358 | 120.852 $\pm$ 0.142 |
> | | 2.336 | 126.607 $\pm$ 0.445 | 2.350 | 126.079 $\pm$ 0.170 |
> | | 2.345 | 131.816 $\pm$ 0.437 | 2.355 | 131.320 $\pm$ 0.179 |
> | | 2.380 | 137.091 $\pm$ 0.550 | 2.333 | 136.545 $\pm$ 0.212 |
> | | 2.405 | 142.317 $\pm$ 0.560 | 2.341 | 141.892 $\pm$ 0.303 |
> | | 2.412 | 147.530 $\pm$ 0.565 | 2.329 | 147.093 $\pm$ 0.312 |
> | | 2.403 | 152.875 $\pm$ 0.529 | 2.334 | 152.377 $\pm$ 0.257 |
> | | 2.351 | 158.193 $\pm$ 0.515 | 2.321 | 157.651 $\pm$ 0.253 |
> | | 2.379 | 163.457 $\pm$ 0.480 | 2.314 | 163.083 $\pm$ 0.233 |
> ---
>
> ## Final Remarks
>
> We thank the reviewer again for the detailed and insightful feedback. In response, we will:
>
> - Improve clarity by reorganizing and annotating technical content;
> - Provide intuitive explanations for key lemmas and parameters;
> - Clarify and strengthen comparisons to related work;
> - Expand experimental analysis and reporting.
>
> We hope these clarifications demonstrate the rigor and value of our contributions.

---

> > ### Comment · Reviewer_bRp6 · 2025-08-06
> > **Rebutal comments**
> >
> > Thanks for your rebuttal that clarified most of my questions.
> > It seems to me that the comparison with Yang et al (Q4 above) is one of the central points of the paper, reflecting its contribution = achieving **similar** performance with first-order (instead of second) information+relaxed smoothness. - the writing might be improved to make it clearer.
> > Generally, there is quite a lot of work to be done on the writing of the paper to make it clearer and incorporate all the reviewers comments. But I think this can be done and given the clarifications of the rebuttal I will increase my score but I still think the paper is borderline.

---

> ### Author Response · Authors · 2025-08-06
>
> Thank you very much for your thoughtful follow-up and for considering an increased score based on our clarifications.
>
> We truly appreciate your recognition that our contribution lies in achieving competitive performance using only first-order information under relaxed smoothness assumptions, while employing a distinct restart condition to enable acceleration. We are also grateful for your constructive suggestion on improving the writing. We will take this seriously and are fully committed to thoroughly revising the manuscript to improve clarity and incorporate all the reviewers’ comments.
>
> Thank you again for your time and valuable feedback.

---

### Official Review · Reviewer_3FGu · 2025-07-03

**Clarity:** 2
**Significance:** 2
**Originality:** 2
**Rating:** 4
**Confidence:** 3

**Summary:**

This work proposes an accelerated bi-level first-order optimization method for problems where the upper-level objective may be non-convex and the lower-level objective is $\mu$-strongly convex. Building on Nesterov’s acceleration framework, the authors achieve an improved iteration complexity of $O(\epsilon^{-7/4})$ compared to the standard $O(\epsilon^{-2})$ rate. The theoretical analysis focuses on deterministic settings and assumes first-order or second-order Hölder/Lipschitz continuity. Additionally, the paper presents numerical experiments on the MNIST and 20 Newsgroups datasets over the proposed method in comparison to existing approaches.

**Questions:**

Algorithm 1 includes local iterations for the AGD steps on the order of $O(T)$, as well as periodic restart steps, where each pair of successive restarts constitutes one epoch. Could the authors clarify how the total number of iterations is counted to achieve the claimed overall iteration complexity of $O(\epsilon^{-7/4})$?

**Ethical Concerns:**

["NO or VERY MINOR ethics concerns only"]

**Final Justification:**

The paper presents a first-order method for solving complex bi-level optimization problems with improved time complexity. The authors have addressed my concerns, and my overall assessment remains unchanged.

**Limitations:**

yes

**Quality:**

2

**Strengths And Weaknesses:**

**Strengths**

* The paper presents an efficient, first-order accelerated method with an iteration complexity of $O(\epsilon^{-7/4})$ for a bi-level optimization problem in which the upper-level objective is non-convex and the lower-level objective is strongly convex, which can be a challenging setting.
* The authors employ more relaxed assumptions of Hölder continuity instead of Lipschitz continuity for certain second-order conditions. The second-order Lipschitz continuity assumptions appear to be standard in this literature.


**Weaknesses**

* $\kappa$ represents the ratio between the maximum smoothness constants for the first- and second-order bounds and the strong convexity parameter, and it can be considerably large; thus, comparing this result to the accelerated rate in [1] can be significant. Also, at [1], $\kappa$ is defined as the conditional number of the lower function, while here the smoothness bound $\mathcal{l}$ bounds all the first and second order Hölder/Lipschitz bounds.
* The analysis only applies to the deterministic case.
* The parameters used in Algorithm 1, such as the smoothness parameters, are generally unknown and difficult to tune.
* The empirical results should be run over several seeds and presented with error bars.



[1] Yang, Haikuo, et al. "Accelerating inexact hypergradient descent for bilevel optimization." arXiv preprint arXiv:2307.00126 (2023).

---

> ### Author Rebuttal · Authors · 2025-07-31
>
> We thank Reviewer 3FGu for their thoughtful review and for recognizing the significance of our contributions, notably:
>
> - The development of an efficient accelerated first-order method for the challenging nonconvex–strongly-convex bilevel optimization setting;
> - The use of **Hölder continuity**, which generalizes standard Lipschitz assumptions and broadens applicability;
> - The improved iteration complexity of $O(\epsilon^{-7/4})$ under first-order oracle access, matching or surpassing the performance of prior second-order oracles.
>
> We appreciate the constructive feedback, and we respond to the key concerns in detail below.
>
>
> ---
>
> W1. On the use and interpretation of the condition number
> $\kappa$
> Thank you for raising this clarification. Indeed, our definition of $\kappa$ encompasses all the relevant smoothness constants ($\ell := \max\{C_f, L_f, \rho_f, H_f, L_g, \rho_g, M_g\}$), including first- and second-order Hölder/Lipschitz parameters for both $f$ and $g$, similar definition as in [1]. While in [2], $\kappa$ is defined as $\max{L_g, L_f}/\mu$. In [2, Lemma 2.6] and [2, Lemma 2.7], the detailed form of Lipschitz constant is actually dependent on second order or third order Lipschitz bounds. The authors of [2] omit the high-order smoothness constants and only consider $\{L_f,L_g,\mu\}$. Thus under our definition of $\kappa$, the complexity bound of [2] in our paper's  Table 1 should be modified as following:
>
> |  |  Gc($f$, $\epsilon$) | Gc($g$, $\epsilon$) | JV($g$, $\epsilon$) | HV($g$, $\epsilon$)|
> |---------|---------|---------|---------|---------|
> | RAHGD([2]) | $\tilde{\mathcal{O}}(\ell^{3/4}\kappa^{11/4} \epsilon^{-7/4})$ | $\tilde{\mathcal{O}}(\ell^{3/4}\kappa^{13/4} \epsilon^{-7/4})$ | $\tilde{\mathcal{O}}(\ell^{3/4}\kappa^{11/4} \epsilon^{-7/4})$ |  $ \tilde{\mathcal{O}}(\ell^{3/4}\kappa^{13/4} \epsilon^{-7/4})$|
> This will ensure transparency and consistency in how $\kappa$ is used across works.
>
>
> ---
>
> W2. On applicability to only deterministic settings
>
> We agree that extending the method to the stochastic setting is an important direction. However, our focus in this work is to first close the gap in deterministic nonconvex–strongly convex bilevel optimization using fully first order oracle under Hölder continuity. Our current analysis lays the necessary groundwork for future stochastic extensions. [5] proposes the Moving-Average Stochastic Bilevel Optimization (MA-SOBA) framework, which achieves an optimal sample complexity of
>  under relaxed smoothness assumptions. [6] studies the complexity of first-order methods in stochastic bilevel optimization, providing tight upper and lower bounds under different assumptions. We will add a discussion on this in the limitations section to better reflect this scope.
>
>
> ---
>
> W3. On unknown parameters (e.g., smoothness constants) in Algorithm 1
>
> This is a valid concern. In our current theoretical analysis, we assume known smoothness constants to simplify exposition and achieve clean complexity bounds. In practice, we use a grid search over reasonable ranges (see Section 5).  Our parameters setting of Algorithm 1 is common in work before [1,2].
>
>
> ---
>
> W4. On empirical results and reproducibility (seeds and error bars)
> We acknowledge this oversight.   We provide experiments on Data Cleaning to report the mean ± standard deviation over 5 random seeds. Due to space limitations, we only report results for "RAGD-GS" and "RAHGD".
>
> | Algorithm | p =0.2 (train loss) | p =  0.2 (Running Time, mean $\pm$ std) | p = 0.4 (train loss) | p =0.4 (Running Time, mean $\pm$ std)) |
> |---|---|---|---|---|
> | RAGD-GS |19.035| 0$\pm$0|19.035| 0$\pm$0|
> || 2.481 | 7.267 $\pm$ 0.300 | 2.498 | 7.530 $\pm$ 0.329 |
> | | 2.370 | 14.795 $\pm$ 0.737 | 2.365 | 15.142 $\pm$ 0.726 |
> | | 2.363 | 22.279 $\pm$ 1.043 | 2.360 | 22.813 $\pm$ 1.150 |
> | | 2.326 | 29.757 $\pm$ 1.425 | 2.348 | 30.506 $\pm$ 1.559 |
> | | 2.352 | 37.322 $\pm$ 1.857 | 2.336 | 38.144 $\pm$ 1.910 |
> | | 2.340 | 44.875 $\pm$ 2.221 | 2.312 | 45.774 $\pm$ 2.293 |
> | | 2.379 | 52.352 $\pm$ 2.607 | 2.324 | 53.355 $\pm$ 2.612 |
> | | 2.313 | 59.811 $\pm$ 2.929 | 2.355 | 61.086 $\pm$ 2.952 |
> | | 2.345 | 67.254 $\pm$ 3.304 | 2.318 | 68.674 $\pm$ 3.317 |
> | | 2.320 | 74.777 $\pm$ 3.669 | 2.344 | 76.266 $\pm$ 3.668 |
> | | 2.345 | 82.236 $\pm$ 4.048 | 2.311 | 83.947 $\pm$ 4.028 |
> | | 2.363 | 89.775 $\pm$ 4.380 | 2.325 | 91.692 $\pm$ 4.317 |
> | | 2.319 | 97.373 $\pm$ 4.677 | 2.342 | 99.348 $\pm$ 4.586 |
> | | 2.343 | 105.049 $\pm$ 5.002 | 2.318 | 106.943 $\pm$ 4.939 |
> | | 2.339 | 112.495 $\pm$ 5.129 | 2.334 | 114.558 $\pm$ 5.286 |
> | | 2.357 | 120.020 $\pm$ 5.300 | 2.332 | 122.203 $\pm$ 5.693 |
> | | 2.353 | 127.511 $\pm$ 5.410 | 2.318 | 129.776 $\pm$ 5.982 |
> | | 2.340 | 135.009 $\pm$ 5.479 | 2.323 | 137.505 $\pm$ 6.313 |
> | | 2.333 | 142.509 $\pm$ 5.575 | 2.326 | 145.200 $\pm$ 6.711 |
> | | 2.306 | 149.887 $\pm$ 5.774 | 2.283 | 152.876 $\pm$ 7.043 |
> | | 2.315 | 157.318 $\pm$ 6.016 | 2.339 | 160.518 $\pm$ 7.363 |
> | RAHGD |19.035| 0$\pm$0|19.035| 0$\pm$0|
> || 3.215 | 5.238 $\pm$ 0.054 | 3.277 | 5.312 $\pm$ 0.066 |
> | | 2.469 | 10.463 $\pm$ 0.083 | 2.436 | 10.502 $\pm$ 0.112 |
> | | 2.404 | 15.692 $\pm$ 0.081 | 2.383 | 15.714 $\pm$ 0.140 |
> | | 2.350 | 20.893 $\pm$ 0.090 | 2.361 | 20.946 $\pm$ 0.158 |
> | | 2.385 | 26.244 $\pm$ 0.063 | 2.355 | 26.148 $\pm$ 0.168 |
> | | 2.375 | 31.493 $\pm$ 0.064 | 2.325 | 31.407 $\pm$ 0.227 |
> | | 2.417 | 36.757 $\pm$ 0.118 | 2.331 | 36.689 $\pm$ 0.227 |
> | | 2.345 | 42.022 $\pm$ 0.163 | 2.370 | 41.977 $\pm$ 0.249 |
> | | 2.375 | 47.243 $\pm$ 0.223 | 2.330 | 47.183 $\pm$ 0.297 |
> | | 2.345 | 52.511 $\pm$ 0.273 | 2.365 | 52.374 $\pm$ 0.305 |
> | | 2.375 | 57.832 $\pm$ 0.327 | 2.328 | 57.587 $\pm$ 0.325 |
> | | 2.403 | 63.160 $\pm$ 0.355 | 2.339 | 62.804 $\pm$ 0.307 |
> | | 2.351 | 68.480 $\pm$ 0.519 | 2.362 | 68.005 $\pm$ 0.295 |
> | | 2.383 | 73.844 $\pm$ 0.474 | 2.338 | 73.289 $\pm$ 0.269 |
> | | 2.376 | 79.162 $\pm$ 0.513 | 2.351 | 78.672 $\pm$ 0.209 |
> | | 2.387 | 84.456 $\pm$ 0.480 | 2.352 | 83.915 $\pm$ 0.221 |
> | | 2.386 | 89.692 $\pm$ 0.506 | 2.340 | 89.101 $\pm$ 0.244 |
> | | 2.374 | 94.928 $\pm$ 0.424 | 2.339 | 94.354 $\pm$ 0.205 |
> | | 2.367 | 100.269 $\pm$ 0.291 | 2.339 | 99.634 $\pm$ 0.239 |
> | | 2.342 | 105.533 $\pm$ 0.295 | 2.292 | 105.029 $\pm$ 0.179 |
> | | 2.348 | 110.828 $\pm$ 0.272 | 2.360 | 110.262 $\pm$ 0.149 |
> | | 2.394 | 116.088 $\pm$ 0.272 | 2.339 | 115.506 $\pm$ 0.141 |
> | | 2.376 | 121.354 $\pm$ 0.284 | 2.358 | 120.852 $\pm$ 0.142 |
> | | 2.336 | 126.607 $\pm$ 0.445 | 2.350 | 126.079 $\pm$ 0.170 |
> | | 2.345 | 131.816 $\pm$ 0.437 | 2.355 | 131.320 $\pm$ 0.179 |
> | | 2.380 | 137.091 $\pm$ 0.550 | 2.333 | 136.545 $\pm$ 0.212 |
> | | 2.405 | 142.317 $\pm$ 0.560 | 2.341 | 141.892 $\pm$ 0.303 |
> | | 2.412 | 147.530 $\pm$ 0.565 | 2.329 | 147.093 $\pm$ 0.312 |
> | | 2.403 | 152.875 $\pm$ 0.529 | 2.334 | 152.377 $\pm$ 0.257 |
> | | 2.351 | 158.193 $\pm$ 0.515 | 2.321 | 157.651 $\pm$ 0.253 |
> | | 2.379 | 163.457 $\pm$ 0.480 | 2.314 | 163.083 $\pm$ 0.233 |
> ---
>
> ---
>
> ### Q1. On Complexity Accounting: AGD Steps and Overall Iteration Complexity
>
> Thank you for the question. In our paper, we denote the total number of epochs as $T$, where each epoch corresponds to one outer iteration of Algorithm 1 before an $\mathcal{O}(\epsilon)$-stationary point is found.
>
> In the proof of Proposition 1, we assume that the algorithm has not yet found an $\mathcal{O}(\epsilon)$-stationary point after epochs $1$ to $t$. Let us denote by $k$ the total number of outer iterations at the end of epoch $t$. As shown in line 505, we derive the following descent guarantee for the $t$-th epoch:
>
> $$
> L_\lambda^*(x_0) - L_\lambda^*(x_k) \geq \frac{1}{100} L^{-\frac{1}{2}} \left(\frac{64\epsilon}{64c+1}\right)^{\frac{4 + 3\nu_f}{2 + 2\nu_f}} \cdot k \cdot H_\nu^{-\frac{1}{2 + 2\nu_f}}. \tag{1}
> $$
>
> By summing inequality (1) over all $t$ epochs, we obtain:
>
> $$
> \Delta_\lambda \geq L_\lambda^*(x_{0,0}) - L_\lambda^*(x_{t,k}) \geq \frac{1}{100} L^{-\frac{1}{2}} \left(\frac{64\epsilon}{64c+1}\right)^{\frac{4 + 3\nu_f}{2 + 2\nu_f}} \cdot K \cdot H_\nu^{-\frac{1}{2 + 2\nu_f}}, \tag{2}
> $$
>
> where $K$ is the total number of outer iterations. Since $\Delta_\lambda$ is finite, it follows that $K$ must also be finite, and therefore $t$ must be bounded. This implies that Algorithm 1 terminates in a finite number of outer iterations.
>
> The resulting outer-loop complexity of $\widetilde{\mathcal{O}}(\epsilon^{-7/4})$ matches the known optimal complexity for nonconvex optimization [4]. Moreover, as established in Theorem 2, we provide the total first-order oracle complexity, which includes the cost of inner AGD iterations. The inner AGD subroutine enjoys linear convergence, so its per-call complexity is logarithmic in $1/\epsilon$.
>
> We will revise the main text and appendix to better clarify this complexity accounting and ensure full transparency.
>
> ---
>
> **Reference**
> [1] Lesi Chen, Yaohua Ma, and Jingzhao Zhang. "Near-optimal nonconvex-strongly-convex bilevel optimization with fully first-order oracles." arXiv preprint arXiv:2306.14853 (2023).
> [2] Yang, Haikuo, et al. "Accelerating inexact hypergradient descent for bilevel optimization." arXiv preprint arXiv:2307.00126 (2023).
> [3] Jeongyeol Kwon, Dohyun Kwon, Stephen Wright, and Robert D Nowak. "A fully first-order method  for stochastic bilevel optimization". In International Conference on Machine Learning, pages  18083–18113. PMLR (2023).
> [4] Huan Li and Zhouchen Lin. "Restarted nonconvex accelerated gradient descent: No more polylogarithmic factor in the in the o (epsilonˆ(-7/4)) complexity". Journal of Machine Learning Research,  24(157):1–37 (2023).
> [5] Chen, Xuxing, Tesi Xiao, and Krishnakumar Balasubramanian. "Optimal algorithms for stochastic bilevel optimization under relaxed smoothness conditions." Journal of Machine Learning Research 25, no. 151 (2024): 1-51.
> [6] Kwon, Jeongyeol, Dohyun Kwon, and Hanbaek Lyu. "On the complexity of first-order methods in stochastic bilevel optimization." arXiv preprint arXiv:2402.07101 (2024).

---

> > ### Comment · Reviewer_3FGu · 2025-08-06
> >
> > Thank you for the comments. Could you please re-upload the mathematical equations, as they are currently difficult to read in the format provided?

---

> > > ### Author Response · Authors · 2025-08-07
> > >
> > > Thank you very much for your continued engagement and for pointing this out.
> > >
> > > We apologize for the formatting issues in our previous response. We have now re-uploaded the mathematical equations in a cleaner and more readable format, ensuring they render correctly.
> > >
> > > The equations are as follows:
> > >
> > > $$ L\_\lambda^* (x_0) - L_\lambda^* (x_k) \geq \frac{1}{100} L^{-\frac{1}{2}} \left(\frac{64\epsilon}{64c+1}\right)^{\frac{4 + 3\nu_f}{2 + 2\nu_f}} k  H_\nu^{-\frac{1}{2 + 2\nu_f}}.\tag{1}$$
> > >
> > > By summing Equation (1) over all
> > >  epochs, we obtain:
> > >
> > > $$ \Delta_\lambda \geq L_\lambda^* (x_{0,0}) - L_\lambda^* (x_{t,k}) \geq \frac{1}{100} L^{-\frac{1}{2}} \left(\frac{64\epsilon}{64c+1}\right)^{\frac{4 + 3\nu_f}{2 + 2\nu_f}}  K H_\nu^{-\frac{1}{2 + 2\nu_f}}, \tag{2} $$
> > >
> > > Please let us know if there are any remaining issues or if further clarification is needed—we’d be happy to assist.
> > >
> > > We greatly appreciate your time and feedback.

---

> > > > ### Comment · Reviewer_3FGu · 2025-08-07
> > > >
> > > > I appreciate the author's clarifications, which have addressed my concerns.

---

### Note · Authors · 2025-08-15

We sincerely thank all reviewers for their constructive feedback and engagement during the rebuttal and discussion phase. Your comments have greatly improved the clarity and completeness of our work. No major errors were identified; comments mainly concerned clarifying definitions, adding experimental results, and minor textual improvements, all of which were addressed in our rebuttal.

Reviewers 3FGu, bRp6, and Kp5u expressed support, with questions on clarity, experiments, advantages, and the motivation for the Hölder continuity assumption. In our rebuttal, we showed that:
1. Our assumptions are reasonable and broadly applicable.
2. We extend first-order acceleration to bilevel optimization under general smoothness via a restart condition, guaranteeing convergence to an $\epsilon$-stationary point without Hessian–vector products.
3. We added empirical results with error bars and extra gradient-norm experiments.

After addressing these concerns, we appreciate that reviewers raised their ratings or maintained positive scores.

Reviewer aCYi—along with Reviewer 3FGu—raised interest in extending our method to stochastic settings. We detailed two key challenges:
1. Restart trigger noise — Deterministic restart criteria (Eq. (5)) become unreliable under stochasticity.
2. Gradient precision — Achieving $\sigma \propto \epsilon$ in stochastic bilevel problems may require large inner-loop sample sizes, impacting complexity.

While Fang et al. (2019) gave a sharp analysis for single-level stochastic optimization under Lipschitz continuity, their techniques lack momentum-based acceleration and cannot be directly extended to stochastic bilevel, where upper- and lower-level noise interactions require different analytical tools. We focused on building a rigorous deterministic foundation here, leaving the stochastic extension for future work.

In summary, this work presents the first accelerated, fully first-order method for nonconvex–strongly-convex bilevel optimization under general Hölder continuity, achieving $\tilde{\mathcal{O}}(\epsilon^{-7/4})$ first-order oracle complexity in the Lipschitz case.

All concerns have been addressed, and the maintained or improved scores reflect the novelty, correctness, and applicability of our work. In the revision, we will further highlight our advantages and provide more explanations and justifications for the assumptions. We thank all reviewers and the AC for their time and valuable input.

---

### Decision · Program_Chairs · 2025-09-17

**Decision:**

Reject

**Comment:**

The paper studies bilevel optimization and propose an accelerated framework under the Holder continuity condition.  The main weaknesses lies in the novelty of the algorithm design and the restriction of the analysis to the deterministic setting, given that most bilevel algorithms are designed for the general stochastic setting. These concerns remain after rebuttal and discussion. I recommend reject.